# Gut Microbiome Modulation in Hepatocellular Carcinoma: Preventive Role in NAFLD/NASH Progression and Potential Applications in Immunotherapy-Based Strategies

**DOI:** 10.3390/cells14020084

**Published:** 2025-01-09

**Authors:** Elisa Monti, Clara Vianello, Ilaria Leoni, Giuseppe Galvani, Annalisa Lippolis, Federica D’Amico, Sara Roggiani, Claudio Stefanelli, Silvia Turroni, Francesca Fornari

**Affiliations:** 1Department for Life Quality Studies, University of Bologna, Corso d’Augusto 237, 47921 Rimini, Italy; elisa.monti10@unibo.it (E.M.); clara.vianello2@unibo.it (C.V.); ilaria.leoni5@unibo.it (I.L.); giuseppe.galvani2@unibo.it (G.G.); annalisa.lippolis@studio.unibo.it (A.L.); claudio.stefanelli@unibo.it (C.S.); 2Centre for Applied Biomedical Research—CRBA, University of Bologna, 40138 Bologna, Italy; 3Unit of Microbiome Science and Biotechnology, Department of Pharmacy and Biotechnology, University of Bologna, 40126 Bologna, Italy; federica.damico8@unibo.it (F.D.); sara.roggiani3@unibo.it (S.R.); silvia.turroni@unibo.it (S.T.); 4Human Microbiomics Unit, Department of Medical and Surgical Sciences, University of Bologna, 40138 Bologna, Italy; 5IRCCS Azienda Ospedaliero, Universitaria di Bologna, 40138 Bologna, Italy

**Keywords:** HCC, NAFLD, microbiome, immunotherapy

## Abstract

Hepatocellular carcinoma (HCC) is a heterogeneous tumor associated with several risk factors, with non-alcoholic fatty liver disease (NAFLD) emerging as an important cause of liver tumorigenesis. Due to the obesity epidemics, the occurrence of NAFLD has significantly increased with nearly 30% prevalence worldwide. HCC often arises in the background of chronic liver disease (CLD), such as nonalcoholic steatohepatitis (NASH) and cirrhosis. Gut microbiome (GM) alterations have been linked to NAFLD progression and HCC development, with several investigations reporting a crucial role for the gut–liver axis and microbial metabolites in promoting CLD. Moreover, the GM affects liver homeostasis, energy status, and the immune microenvironment, influencing the response to immunotherapy with interesting therapeutic implications. In this review, we summarize the main changes in the GM and derived metabolites (e.g., short-chain fatty acids and bile acids) occurring in HCC patients and influencing NAFLD progression, emphasizing their potential as early diagnostic biomarkers and prognostic tools. We discuss the weight loss effects of diet-based interventions and healthy lifestyles for the treatment of NAFLD patients, highlighting their impact on the restoration of the intestinal barrier and GM structure. We also describe encouraging preclinical findings on the modulation of GM to improve liver functions in CLD, boost the antitumor immune response (e.g., probiotic supplementations or anti-hypercholesterolemic drug treatment), and ultimately delay NAFLD progression to HCC. The development of safe and effective strategies that target the gut–liver axis holds promise for liver cancer prevention and treatment, especially if personalized options will be considered.

## 1. Introduction

Hepatocellular carcinoma (HCC) is a prevalent type of primary liver cancer, representing 80–90% of cases [1]. HCC is the sixth most common cancer worldwide and the third leading cause of cancer-related mortality in 2020, with an estimated 5-year survival rate of 20%. Although cirrhosis represents the predominant risk factor for HCC, a profound epidemiological shift in the incidence of HCC from patients with virus-related liver disease to those with non-viral etiologies has been observed. Because of implemented vaccination programs and improved treatment options for hepatitis B virus (HBV) and hepatitis C virus (HCV) infections, non-alcoholic fatty liver disease (NAFLD) is emerging as an important cause of HCC development. Moreover, due to the obesity epidemics, the occurrence of NAFLD has significantly increased in the last two decades, with an estimated overall prevalence ranging from 25% to 30% worldwide [2]. NAFLD, whose denomination has been recently changed to “metabolic dysfunction-associated steatotic liver disease (MASLD)”, represents the highest rapidly increasing risk factor for HCC [3]. Projections from mathematical models suggest an increase in cases of advanced liver disease and liver-related mortality in the coming years, especially if actions are not put in place to improve NAFLD and non-alcoholic steatohepatitis (NASH) diagnosis and increase public awareness to promote the adoption of healthier lifestyles [4].

The therapeutic landscape of advanced HCC has witnessed a radical change in recent years, with five tyrosine kinase inhibitors (TKIs) approved across first- and second-line settings and two immunotherapies in the first setting [5]. Over the past decade, immuno-oncology has represented a paradigm shift in the treatment of several kinds of deadly cancers, including HCC, showing unprecedented clinical responses, rapid drug development, and accelerated approvals [6]. In particular, two immunotherapy-based strategies with atezolizumab (anti Programmed Death-Ligand 1—PD-L1) plus bevacizumab (anti Vascular Endothelial Growth Factor—VEGF) and durvalumab (anti-PD-L1) plus tremelimumab (anti Cytotoxic T-Lymphocyte Antigen 4—CTLA-4) are now recognized as standard of care for advanced HCC [7]. Despite the impressive improvement in terms of overall survival (OS) and objective response rate (ORR), which is defined as the percentage of patients who achieve a complete or partial response, not all patients are eligible for immunotherapy; furthermore, patients with NASH-driven HCC who received immune checkpoint inhibitors (ICIs) showed reduced OS compared to patients with viral etiologies [8]. Research efforts are now focused on identifying predictive biomarkers of responses to stratify patients and new therapeutic targets for combined and tailored treatments.

Unbalances in the gut microbiome (GM), or dysbiosis, are known to affect liver homeostasis and are emerging as a contributing factor to the onset and progression of several liver diseases [9]. Although the liver is not in direct contact with the GM, there is a tight anatomic link between them as well as an intimate bi-directional relationship and metabolite exchange known as the gut–liver axis [10]. Several studies have profiled dysbiosis in chronic liver diseases, such as non-alcoholic steatohepatitis (NASH) and cirrhosis, highlighting that specific microbiome signatures can distinguish HCC from non-HCC cases and could be exploited for early diagnosis [11]. Further evidence also supports the critical role of GM-derived metabolites, such as secondary bile acids (BAs), in influencing the gut permeability and the bacterial translocation to the liver [12], leading to an altered tumor microenvironment and immunological response, as well as metabolic pathways associated with HCC development and treatment response [13].

This review will summarize the alterations of the GM in chronic liver diseases, particularly in NAFLD/NASH pathogenesis and progression to HCC and describe the potential of GM-based treatments to enhance the effect of ICIs in the perspective of increasingly personalized precision medicine. Given the role of the GM in modulating response or toxicity to cancer treatments by interfering with host immunity [14], strategies to manipulate the microbiome represent a future challenge to improve the efficacy of immunotherapy in less responsive HCC subgroups.

## 2. Direct and Indirect Roles of Gut Microbiota in HCC Development

The modulation of HCC development by the GM is a dynamic and reversible process. HCCs resulting from different etiologies are characterized by peculiar GM profiles that share a marked decline in alpha diversity and a shift in composition [15]. The liver receives more than 70% of its blood supply from the portal vein; specifically, through enterohepatic circulation, the liver can receive several molecules from the GM, such as microbe-associated molecular patterns (MAMPs) and metabolites, which play an important role in hepatocarcinogenesis [16]. In fact, with its multiple metabolic and immunoregulatory functions, the liver is engaged in extensive cross-communication with the gut via enterohepatic circulation [17]. Here, we will delineate the direct and indirect mechanisms through which gut dysbiosis influences HCC progression, emphasizing how alterations in the permeability of the intestinal epithelial barrier can facilitate the translocation of microorganisms and MAMPs into the liver and describing how microbial metabolites interfere with liver homeostasis and metabolic functions (Figure 1).

### 2.1. Gut Integrity in Chronic Liver Disease and HCC

The intestinal barrier plays a vital role in absorbing essential nutrients and preventing the entry of microorganisms from the gut into systemic circulation. Disruption of the integrity of the intestinal barrier alters its permeability by promoting the translocation of microorganisms and derived molecules from the intestinal lumen to the liver via portal circulation, thereby inducing an inflammatory state and subsequent liver damage and fibrosis, as reviewed elsewhere [18]. A central dogma of host–microbial maladaptation is the “leaky gut” model, which is characterized by a reduced microbe-impermeable inner mucus layer. In this regard, the study by Ijsennager et al. showed that the ablation of farnesoid X receptor (Fxr), which is a master regulator of BA homeostasis, greatly impacting gene expression in colon epithelial cells and increasing the mucus barrier, proving that liver-to-gut communication is crucial for the intestinal health [19]. In addition, Everald et al. demonstrated that administration of viable, but not heat-killed, *Akkermansia muciniphila* improved the mucus layer thickness by increasing the expression of the antimicrobial peptide Reg3γ in colon epithelial cells of mice undergoing a high-fat diet (HFD) regimen [20], providing a rationale for the development of probiotic formulations for obesity-related disorders, such as NAFLD. Indeed, intestinal epithelial barrier disruption is a crucial element in the pathophysiology of MASLD, contributing to the promotion of intra and extrahepatic damage, changes in tight junctions (TJs), increased intestinal permeability, and dysbiosis [21]. Transmembrane proteins such as claudins, occludin, and junctional adhesion molecules are TJ family members, which interact with the zona occludens (ZO) scaffolding proteins. Mouries and colleagues reported a reduction in tight junction protein zonulin-1 (ZO-1) and an increase in plasmalemma vesicle-associated protein 1 (PV1), a marker of gut vascular barrier disruption, in HFD-treated mice, highlighting the disruption of these two barriers as an early event in NAFLD-to-NASH progression [22].

A growing body of research has proposed a connection between gut epithelial barrier function, GM composition, and HCC development. In fact, inflammation and gut dysbiosis can cause liver damage due to translocation of microbes and/or pro-inflammatory molecules to the liver, where they can exert pro-tumorigenic effects [23]. Dapito et al. showed that the gut–liver axis is impaired in animal models of chronic liver injury, where it promotes hepatocarcinogenesis by translocation of microbial components termed pathogen-associated molecular patterns (PAMPs) that trigger inflammatory responses by activating Toll-like receptors (TLRs) [24]. In line with this, Ram et al. [25] reported the effects of the terpenoid-based bioactive compound, Nimbolide, on gut dysbiosis in a diethylnitrosamine (DEN)-HCC murine model, showing partial restoration of gut eubiosis and prevention of microbial translocation by enhancing intestinal barrier integrity (increased TJ proteins) and alleviating inflammation, as demonstrated by the downregulation of inflammation-associated proteins (TLR4, MyD88, NF-kBp65, TNF-alpha, IL-6). Concerning GM composition, a higher prevalence of opportunistic bacteria was found in HCC mice, reflecting the inflammatory state associated with lipopolysaccharide (LPS) accumulation. Nimbolide decreased opportunistic bacteria, such as *Escherichia coli* and *Bacteroides* spp., which are more likely to translocate to the liver through a permeable intestine. On the other hand, Nimbolide increased beneficial bacteria from the *Bifidobacterium* and *Lactobacillus* genera, which can limit the colonization and excessive growth of enteropathogens, thereby reducing the risk of infection. Similarly, Zhang et al. reported increased serum endotoxin (e.g., LPS) levels in cirrhotic and HCC patients and in a DEN-induced HCC rat model, which also showed an imbalance in GM subpopulations, including suppression of *Lactobacillus*, *Bifidobacterium*, and *Enterococcus* species, an increase in intestinal inflammation, and alterations in microvilli structure [26]. Daily treatment with a commercial probiotic formulation (VSL#3—VSL Pharmaceutical) restored gut dysbiosis and mucosal barrier function, decreased serum LPS, and impaired HCC development, suggesting that probiotics administration could have preventive effects in cirrhotic patients, even though further preclinical findings in different models of cirrhosis-HCC would be necessary before translating data into clinics. A recent study revealed that *Klebsiella pneumoniae* is enriched in patients with HCC and showed that mice transplanted with the HCC microbiome present gut barrier injury and translocation of live bacteria to the liver [27]. Mechanistically, a surface protein of *K. pneumoniae* interacts with TLR4 on HCC cells, leading to the activation of oncogenic pathways, highlighting the critical role for an altered GM in hepatocarcinogenesis.

In vivo models are valuable tools to study gut–vascular barrier integrity. One experimental method consists of the injection of fluorescent dextran into the mouse colon to evaluate the presence of fluorescence in biological specimens, such as the plasma, liver, and spleen. Additional techniques include the detection of LPS in plasma samples or immunofluorescence using an anti-PV1 antibody, which is a marker of endothelial cell permeability [28]. Another example of assessing gut integrity in vivo is the evaluation of the TJ protein Occludin in the ileum mucosa through immunofluorescent staining. Wang et al. observed a progressive impairment of the mucosal barrier along with the aggravation of liver disease, ranging from early MASLD to MASH to severe fibrosis to MASH-HCC, in a diet-induced murine model that mimics long-term exposure to a Western diet [29]. In preclinical models, bacterial translocation can also be assessed by measuring portal blood levels of LPS with a chromogenic test, while gut injury can be determined by measuring portal blood levels of markers of intestinal mucosal damage, such as intestinal fatty acid-binding protein (i-FABP) and diamine oxidase (DAO) by ELISA. Orci et al. described how surgical strategies that prevent/mitigate bacterial translocation by preserving the intestinal barrier after liver resection or transplantation could reduce the recurrence of HCC, inhibiting TLR4 signaling in the liver [30]. Thus, GM restoration, gut barrier integrity, and TLR4 signaling may serve as therapeutic targets for HCC and recurrence prevention in advanced liver disease. In particular, TLR4 is regulated by several noncoding RNAs, including microRNAs (e.g., miR-122, miR-145), and is involved in multidrug resistance [31]. Although promising, microRNA therapeutic strategies have not yet entered the clinical practice due to the early interruption of the first clinical trial in metastatic cancers; this strategy is currently under investigation in clinical trials [32]. Moreover, due to the multitarget nature of microRNAs, caution should be used in proposing this strategy to target TLRs in order to avoid off-target effects.

### 2.2. Dysbiosis in Chronic Liver Disease and HCC

In the last decade, microbiome dysbiosis has emerged as a key factor in the progression of chronic liver diseases (CLDs) and the development of HCC in at-risk populations. Ren et al. investigated, for the first time, the potential of a GM signature for the early diagnosis of HCC in Chinese patient cohorts from different regions of China [33]. Microbial diversity was decreased in cirrhotic patients compared to healthy controls, while it was increased in early HCCs (eHCCs) compared to cirrhosis. Specifically, an enrichment of the Actinobacteria phylum together with the genera *Gemmiger*, *Parabacteroides*, and *Paraprevotella* was reported in eHCCs compared to cirrhosis. A set of 30 operational taxonomic units (OTUs) was identified as an optimal marker to discriminate eHCC from cirrhosis with powerful diagnostic potential (AUC = 80%), showing a similar result to that obtained in colorectal cancer [34]. Ponziani et al. performed a clinical study in NAFLD-related cirrhotic patients with and without HCC to evaluate GM imbalance and inflammatory status in chronic liver diseases [35]. Specifically, fecal levels of calprotectin, a marker of local inflammation released by intestinal neutrophils, were elevated in the HCC group together with plasma levels of proinflammatory chemokines (IL-8, IL-13, CCL3, CCL4, and CCL5) compared to cirrhotic patients without HCC. Microbiome analysis by 16S rRNA amplicon sequencing revealed a higher relative abundance of *Enterococcus* and *Streptococcus*, along with a reduced relative abundance of *Akkermansia* and *Bifidobacterium* in HCC. Notably, *Akkermansia* was the most represented genus in the control group, whereas *Bifidobacterium* was the most represented genus in cirrhotic patients without HCC, highlighting the profound changes occurring in NAFLD-related liver cancer patients. Similarly, Behary and colleagues [36] performed shotgun metagenomic sequencing in NAFLD cirrhotic and NAFLD-HCC patients and reported decreased α-diversity compared to non-NAFLD controls. At the phylum level, NAFLD-HCC was characterized by the expansion of Proteobacteria compared to non-NAFLD controls. At the family level, an increase in *Enterobacteriaceae* was described in NAFLD-HCC compared to both NAFLD-cirrhosis and non-NAFLD controls. A reduction in *Oscillospiraceae* and *Erysipelotrichaceae* was observed in NAFLD-HCC compared to non-NAFLD controls only. At the species level, the NAFLD-HCC microbiome showed enrichment in five bacterial taxa that are well-known short-chain fatty acid (SCFA) producers, namely *Bacteroides xylanisolvens*, *Ruminococcus gnavus*, and *Clostridium bolteae*, which were enriched in NAFLD-HCC and NAFLD-cirrhosis compared to non-NAFLD controls, and *Bacteroides caecimuris* and *Veillonella parvula*, which were increased in NAFLD-HCCs only. As a proof of principle, microbial genes involved in SCFA synthesis, e.g., pyruvate carboxylase, phosphate acetyltransferase, and phosphate butyryltransferase, as well as fecal levels of oxaloacetate and acetylphosphate, which are known SCFA intermediates, were overexpressed in NAFLD-HCC compared to NAFLD-cirrhosis and non-NAFLD, demonstrating HCC specificity that could be exploited for the discovery of non-invasive biomarkers. The NAFLD-HCC GM was also associated with an immunosuppressive milieu characterized by altered expansion of regulatory T cells (Tregs) and decreased CD8+ T cells, paving the way for gut-modulating therapeutic strategies to potentiate immunotherapy in HCC. Another study confirmed the decreased microbiome diversity in cirrhotic and HCC patients compared to NAFLD subjects [37]. A reduced relative abundance of the SCFA-producing bacteria *Blautia* and *Agathobacter* was detected in cirrhotic and HCC patients. Interestingly, an enrichment of fecal bacteria 16S copies in the blood and liver increased progressively from NAFLD to cirrhosis to HCC, showing an enrichment for the *Ruminococcaceae* and *Bacteroidaceae* families and indicating disease-specific bacterial translocation. A significant association between several bacterial genera and metallothionein 1B (MT1B), a scavenger of reactive oxygen species (ROS), was found in liver samples from HCC patients, suggesting that bacterial translocation to the liver, arising from an impaired gut barrier, determines the perturbation of gene expression in the liver, including genes with antioxidant functions which are well-known players during hepatocarcinogenesis. It should be noted that there was a slight discrepancy between studies in terms of microorganisms identified, which could be attributed to several causes, among which are the presence of different etiologic factors, control groups, and microbiome-associated confounding factors [38], as well as methodological differences (for wet and in silico methods). In addition, only one study [33] found that fecal microbial diversity was increased in early HCC in comparison to cirrhosis, while the others reported a reduction in diversity in HCC patients. The authors suggest that the overgrowth of various harmful bacteria or archaea could be detrimental for HCC development in their HBV-related patient cohort, as reported in colorectal carcinoma [39]. A further difference among these studies that might explain that this discrepancy in GM richness in HCC patients represents the different etiologies of patient cohorts, as detailed in Table 1.

A preclinical study by Schneider and coworkers reported the influence of dysbiosis, obtained by *Nlrp6* gene knockout, on the tumor microenvironment in a steatohepatitis-HCC mouse model (NEMO^Δhepa^). The authors demonstrated that bacterial translocation, due to the presence of a leaky intestinal barrier, affected the antitumoral response by expanding monocytic myeloid-derived suppressor cells (mMDSCs) and suppressing CD8+ T cell proliferation [13]. As a proof of principle, antibiotic treatment and *A. muciniphila* administration reduced liver transaminase levels in NEMO^Δhepa/Nlrp6-/-^, mice proving that (1) microbial dysbiosis has detrimental effects on liver injury and (2) *A. muciniphila* supplementation can have a beneficial effect even in the presence of host-derived factors that promote dysbiosis, such as NLRP6 deficiency. Similarly, cirrhotic patients showed a close association between bacterial translocation and activation of fibrogenic and pro-inflammatory pathways that interfere with cancer immunosuppression, highlighting the strict gut–liver crosstalk during disease progression. In line with this, another study reported the overgrowth of fecal *E. coli* in cirrhotic patients with HCC, suggesting its contribution to malignant transformation in cirrhotic patients [40].

Taken together, preclinical and clinical studies have demonstrated the influence of gut dysbiosis in HCC development, pointing out the relevance of microbiome sequencing for biomarker discovery and the potential of manipulating the microbiome as a promising intervention strategy to potentiate the antitumor immune response.

### 2.3. Microbial Metabolites in Chronic Liver Disease and HCC

GM contributes to hepatic energy metabolism through several mechanisms, including the production of SCFAs and the metabolism of BAs, which have been implicated in the progression of HCC [41]. Additionally, epithelial barrier dysfunction, often observed in patients with CLD, may facilitate the translocation of gut metabolites to the liver [42].

For example, the microbial metabolite, 3-(4-hydroxyphenyl)lactate, has been associated with liver fibrosis in NAFLD patients; pathway reconstruction analysis linked it to several species of the human GM belonging to Firmicutes, Bacteroidetes, and Proteobacteria phyla [43]. Serum metabolites have also been investigated in the context of risk stratification in patients with cirrhosis enrolled in screening programs for the early detection of HCC. In the study by Sanchez et al., an analysis of serial blood samples from patients under active surveillance for HCC identified metabolites associated with the risk of HCC development [44]. In particular, an increase in the microbiome-derived metabolite salicyluric glucuronide and a decrease in 3-phenylproprionate and cinnamoylglycine were reported in specimens from cirrhotic patients who developed HCC, which were collected 12 months prior to diagnosis, compared to those who did not. This study highlighted that GM-derived metabolites may be strong candidates for early diagnosis of HCC in at-risk populations, which is a critical step to improve management and reduce mortality in these patients. A huge modification of the GM and serum metabolome has also been reported in HCC compared to cirrhotic patients and healthy individuals [45]. Two key species (*Odoribacter splanchnicus* and *Ruminococcus bicirculans*) and five metabolites (ouabain, taurochenodeoxycholic acid, glycochenodeoxycholate, theophylline, and xanthine) were associated with HCC and outperformed alpha-fetoprotein (AFP) as diagnostic biomarkers.

SCFAs, which are produced by the GM mainly through the fermentation of dietary fiber, exert, in most cases, protective anti-inflammatory effects, enhance barrier integrity, modulate liver metabolism, and even promote anticancer effects, thus counteracting disease progression [46]. For example, butyrate levels were reduced in the plasma of HCC patients compared to healthy individuals [47]. Furthermore, butyrate supplementation was proven to inhibit HCC proliferation and metastasis and improve the therapeutic efficacy of sorafenib by triggering the calcium signaling pathway and increasing ROS production, thus leading to apoptotic cell death in an orthotopic mouse model. Similarly, SCFAs, especially acetate, along with *Lactobacillus reuteri*, were markedly underrepresented in HCC mice [48]. Mechanistically, acetate impairs the production of the proinflammatory cytokine IL-17 by type 3 innate lymphoid cells, which are correlated with poor prognosis in HCC patients. Ma and colleagues reported an increased relative abundance of *Bacteroides thetaiotaomicron* in non-recurrent HCC patients [49]. Acetate mediated its effects in modulating macrophage polarization and activation of cytotoxic CD8+ T cells, exerting antitumor activity in vivo by regulating fatty acid biosynthesis in tumor macrophages. However, other studies have reported contrasting data regarding the effects of SCFAs in HCC tumorigenesis. In HCC animal models, dietary fructose promoted HCC progression through microbial acetate production, responsible for the induction of protein hyper-O-GlcNAcylation in tumor cells [50]. Behary et al. identified marked dysbiosis and increased production of SCFAs in both stool (acetate, butyrate) and serum (butyrate, propionate) specimens from NAFLD-HCC patients compared to NAFLD-cirrhosis and non-NAFLD individuals [36], suggesting a detrimental effect of these metabolites in HCC progression. For butyrate, such a dual role (known as “the butyrate paradox”) has been attributed to its concentration, with higher levels required for histone deacetylase inhibition and, thus, anticancer effects [51]. Further studies are therefore mandatory to clarify the context-specific role of microbial metabolites in hepatocarcinogenesis. Once elucidated, they could serve as diagnostic and prognostic biomarkers in a personalized multi-omics approach.

**Figure 1 cells-14-00084-f001:**
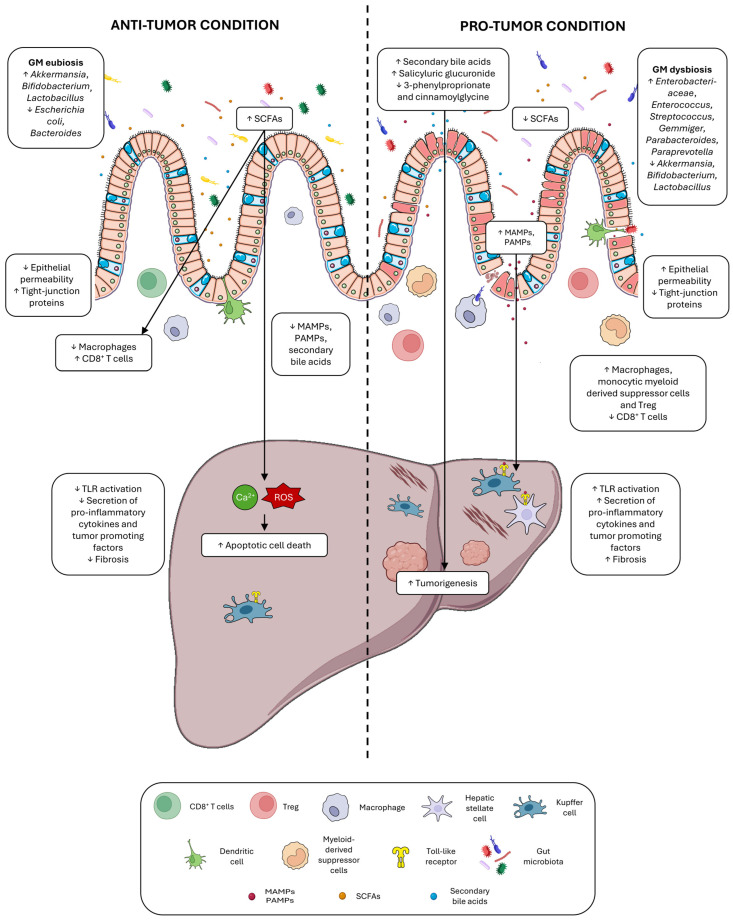
Mechanisms by which the gut microbiome promotes or hinders the progression of HCC. On the left, the mechanisms associated with HCC suppression are characterized by the presence of an eubiotic gut microbiome (GM) profile, enriched in beneficial microbes and capable of producing short-chain fatty acids (SCFAs), which enhance intestinal epithelial integrity and promote apoptosis of tumor cells via reactive oxygen species (ROS), as well as reduced levels of tumor-promoting metabolites such as secondary bile acids and microbe-associated molecular patterns (MAMPs), leading to reduced hepatic inflammation and fibrosis. This environment is characterized by a decrease in inflammatory macrophages and an increase in effector CD8+ T cells, fostering an anti-tumor immune response. On the right, the mechanisms that promote HCC progression are summarized. Gut dysbiosis, reduced ability to produce SCFAs, and increased intestinal permeability (“leaky gut”) may lead to increased generation and translocation of tumor-promoting metabolites, such as secondary bile acids. Dysbiosis also heightens hepatic exposure to gut-derived MAMPs and pathogen-associated molecular patterns (PAMPs), which drive hepatic inflammation, fibrosis, and cell proliferation. This pro-tumor environment is further characterized by an increase in macrophages, monocytic myeloid-derived suppressor cells, and a reduction in CD8+ T cells, contributing to immune evasion. “The figure was partly generated using Servier Medical Art provided by Servier, licensed under a Creative Commons Attribution 4.0 unported license”.

#### Bile Acid Changes in Chronic Liver Disease and HCC

Primary BAs are produced in the liver and then metabolized by specific enzymes of gut microbes to produce secondary BAs, which return to the liver via enterohepatic circulation [52]. BAs play critical roles in the maintenance of a healthy GM, in the homeostasis of lipid and carbohydrate metabolism, as well as in insulin sensitivity and innate immunity. The complexity of enterohepatic circulation, BA metabolism, and signaling has been reviewed elsewhere [53]. In addition to acting as detergents, BAs serve as ligands for several nuclear receptors (e.g., FXR) in the liver and the ileum, activating negative feedback loops that control their production from cholesterol by repressing CYP7A1 expression, the key enzyme in BA synthesis [54]. BA-regulated FXR signaling also interferes with lipid metabolism by repressing sterol regulatory element-binding protein 1c (Srebp1c), the master regulator of de novo lipogenesis in mice, promoting fatty acid oxidation and reducing serum triglyceride levels in humans. Notably, BAs shape the GM, but, in turn, they are metabolized by the intestinal bacterial flora, producing novel BA metabolites and signaling molecules [55]. Indeed, disruption of GM eubiosis can alter the recycling of BAs and has been implicated in the onset of gastrointestinal and extraintestinal tumors, including HCC [56].

Smirnova and coworkers identified changes in fecal BAs derived from microbial metabolism, including increases in 7,12-diketolithocholic acid, glycodeoxycholic acid, and lithocholic acid in NASH compared to NAFLD [57]. Transcriptomic analysis confirmed the deregulation of key microbial genes involved in BA conversion, such as bile salt hydrolase (BSH), which is involved in the first and rate-limiting step of intestinal BA metabolism, and microbial genes belonging to acid-inducible operons (bai), which convert primary to secondary bile acids through 7-dehydroxylation, the main gut modification of BAs. The clinical association of serum deoxycholic acid (DCA) with NASH and active fibrosis and of taurodeoxycholate with cirrhosis decompensation were confirmed in an independent cohort, pointing out the clinical relevance of BAs as biomarkers of disease severity. DCA is a gut microbial metabolite known to cause DNA damage, whose levels were found to be increased in obese individuals [58]. In obesity-related HCC mouse models characterized by diet-induced gut dysbiosis, the enterohepatic circulation of DCA provoked the activation of the ‘senescence-associated secretory phenotype’ in hepatic stellate cells, which in turn secreted pro-inflammatory and tumor-promoting factors in the liver, thereby facilitating HCC development. Similarly, an HFD-induced NASH-HCC mouse model showed accumulation of cholesterol and bile acids in the liver and feces [59]. Antibiotic treatment suppressed tumor development and hepatic accumulation of secondary BAs, which are involved in the activation of the oncogenic mTOR pathway.

BAs also play an important role in immunosurveillance of tumor growth in the liver, where the balance between primary and secondary BAs is fundamental for NKT cell accumulation mediated by the CXCR6/CXCL16 axis [60]. Primary BAs increase CXCL16 expression, favoring a tumor suppressor microenvironment, whereas secondary BAs have the opposite effect. Probiotics, antibiotics, and secondary BA acid inhibitors may be interesting strategies for the development of personalized anticancer therapeutics in primary and metastatic liver tumors. The study by Conde de la Rosa et al. reported the accumulation of BAs in NASH-HCC patients together with the upregulation of steroidogenic acute regulatory protein 1 (STARD1), which stimulates the generation of BAs via the mitochondrial acidic pathway and increases liver tumorigenicity in mouse models [61]. Mechanistically, BAs exert a pro-tumorigenic role in vitro by inducing hepatocyte pluripotency and self-renewal genes. Notably, BAs are host-derived but can shape the microbiome composition through their antimicrobial activity, and in turn, bacterial metabolism can modify the composition of the intestinal and circulating BA pools, thereby interfering with gut and liver homeostasis [62]. Because of the importance of the liver–bile acid–microbiome axis, manipulating the composition of the microbiome or directly the bile acid pool might be a promising strategy to delay the progression of CLD.

## 3. NAFLD-HCC Pathogenesis

NAFLD represents the most prevalent liver disease and is likely to emerge as the leading cause of CLD and HCC development in the near future [63]. The risk of HCC onset is lower in NAFLD patients in comparison to other etiologies, but its high prevalence worldwide impacts a wide number of people, making it a serious global burden for healthcare systems’ policies [64]. NAFLD is a condition characterized by increased lipid content in more than 5% of hepatocytes. Accumulation of triglycerides within hepatocytes (liver steatosis) is a hallmark of NAFLD and is strongly associated with obesity, type 2 diabetes mellitus, and metabolic syndrome [65]. In this regard, experts have reached a consensus that NAFLD does not reflect the current knowledge on the pathology and suggested the update of NAFLD nomenclature to “metabolic associated fatty liver disease—MAFLD”. Its diagnosis should be based on the presence of metabolic dysfunction, such as insulin resistance/type 2 diabetes mellitus, overweight/obesity, and dyslipidemia (low HDL cholesterol and/or hypertriglyceridemia), and not on the absence of other conditions such as alcohol abuse [66]. Parallel to the obesity epidemics, the prevalence of NAFLD is steadily increasing on almost every continent, affecting 30% of the world’s population and being the predominant cause of liver dysfunction worldwide [67].

NAFLD progression is not fully understood, but it is known that a subset of individuals develops NASH, a more serious form of liver damage associated with lobular inflammation and hepatocellular ballooning that can progress to fibrosis through the activation of Kupffer and stellate cells, establishing a vicious cycle of cell injury and death, which prompts compensatory proliferation and disease progression. Non-invasive tests cannot reliably differentiate NAFLD from NASH; however, studies evaluating serum levels of liver enzymes as surrogate biomarkers, despite underestimating the true prevalence, suggest NASH development in about 25% of NAFLD subjects. In some individuals, NASH eventually develop to cirrhosis and/or HCC, which may represent the causes of liver-related death [68].

In the last years, the simplistic and outdated theory of ‘two-hit hypothesis’ has been replaced by the multiple-hit hypothesis, which takes into account multiple synergistically acting events that contribute to the development and progression of NAFLD [69]. These hits include increased intracellular levels of free fatty acids (FFAs) that cause lipotoxicity, adipose tissue alterations leading to the release of hormones (adipokines) and inflammatory chemokines (IL-6 and TNFα), nutritional unbalance, GM dysbiosis, and genetic predisposing factors. All these events contribute to the development of an inflammatory milieu, hepatocyte death, and hepatic stellate cell activation, establishing a negative cycle of tissue regeneration and extracellular matrix deposition. It was thought that fibrosis progression was more frequent in NASH with respect to NAFLD; however, several studies have dismantled this dogma. A dual biopsy cohort study showed that fibrosis progression is observed in 40% of cases irrespective of NAFLD or NASH diagnosis and is associated with diabetes [70]; the degree of fibrosis, and not NASH, is the strongest predictive factor of liver-related or unrelated mortality [71].

Despite the large prevalence of NAFLD in the population, a paradox regarding its evolution highlights that only a small proportion (2.4–12.8%) of cases develops HCC or experiences liver-related death [72]. Notably, over the past decades, several cases of NAFLD-related cirrhosis have been defined as ‘cryptogenic’, masking the real contribution of this etiology to HCC development. It is now well established that NAFLD increases the risk of cirrhosis and HCC and that metabolic traits additionally increase this risk, with diabetes showing the strongest association with HCC, as reported in retrospective studies [73,74]. On the other hand, it is becoming more evident that NAFLD-HCC may occur in the absence of liver cirrhosis in nearly half of cases [75]. From a clinical perspective, it is important to identify both disease progressors and patients to be included in screening programs. The multifactorial basis for this heterogeneity in the clinical course of the disease is under investigation, aiming to identify biological drivers of disease progression and tailored strategies to NAFLD patients.

### 3.1. NAFLD Progression and Dysbiosis of the Gut and Liver Microbiome

Although cirrhosis is the main predisposing factor to malignant transformation of hepatocytes, several other environmental risk factors contribute to disease progression in NAFLD patients. Due to the central role of the gut–liver axis in the pathophysiology of NAFLD-NASH, gut microbial dysbiosis is emerging as a critical event in promoting and exacerbating the fibro-inflammatory response of the liver microenvironment [76]. In this regard, an outstanding study has revealed that specific commensal bacteria, including *Odoribacteraceae*, induce the liver-resident immunosuppressive population of Marco+ macrophages through the production of the metabolite isoallolithocholic acid (isoallo-LCA), thereby suppressing inflammatory responses in periportal vein zones [77]. In particular, this metabolite strongly increased when mice received a strain of *Odoribacteraceae* (*Odori*) while cohoused with specific pathogen-free (SPF) mice, suggesting that the symbiosis of Odori with other bacteria is necessary for the production of isoallo-LCA which, in turn, stimulated the induction of Marco+ Kupffer cells and increased the expression levels of the anti-inflammatory interleukin IL-10. Since gut bacteria, or their related products, can reach the liver through the portal vein, the activation of this macrophage subpopulation is responsible for protecting the liver against harmful inflammatory responses that characterize CLDs. Dysbiosis and intestinal barrier leakage are responsible for the failure of this self-limiting system, promoting hepatic inflammatory disorders such as NASH. This aspect has great therapeutic potential because different microbiome-based strategies can be deployed as primary or secondary prevention, such as antibiotic treatment, prebiotics, probiotics, synbiotics, postbiotics, and fecal microbiota transplantation (FMT). The latter approach was used to investigate whether changes in the GM could alter plasma metabolites and liver DNA methylation in NAFLD patients that underwent three 8-weekly vegan allogenic donor or autologous FMTs [78]. Upon vegan allogenic FMT, the most significant changes were increases in *Eubacterium siraeum*, *Blautia wexlerae*, phenylacetylcarnitine, and phenylacetylglutamine. In addition, metabolites related to glycerophospholipid metabolism and choline derivatives were among the most discriminating between the two groups. Hepatic DNA methylation profiles showed hypomethylation of Threonyl-TRNA Synthetase 1 (TARS) and Zinc finger protein 57 (ZFP57) upon allogenic FMT. Multi-omics analysis identified multiple associations between GM, plasma metabolites, and liver DNA methylation, showing that changes in microbiome composition may have effects beyond the intestine, paving the way for future investigations aimed at developing precision microbiome-based strategies for NAFLD treatment.

On the other hand, host characteristics may influence the selection of specific microbial taxa in the liver, thereby contributing to the modulation of disease biology. For example, Pirola and coworkers demonstrated that host genetics modulate the microbial composition of the liver in NAFLD patients [79]. They focused on variants that positively (PNPLA3-rs738409, TM6SF2-rs58542926, MBOAT7-rs641738) or negatively (HSD17B13-rs72613567) influence the risk of histological severity of NAFLD and on a variant that influences macronutrient preference and carbohydrate intake (FGF21-rs838133). Carriers of the PNPLA3-rs738409 and TM6SF2-rs58542926 risk alleles were enriched in the *Enterobacter* and *Pseudoalteromonas* genera, respectively, whereas the *Lawsonella*, *Prevotella_9*, and *Staphylococcus* genera were enriched in FGF21-rs838133 minor allele carriers, which are linked to sugar consumption. In addition, *Tyzzerella* and *Lactobacillus* exhibited the strongest associations with high (>4 risk alleles) and low (<4 risk alleles) polygenic risk scores (PRS), respectively. PRS also explained the 7.4% variation in genus-level taxa independently of liver steatosis score and obesity, suggesting that host genetics may shape the metabolic liver environment, exerting selective pressure on liver microbial populations. In line with this, a preclinical study [80] reported a close relationship between improved mitochondrial activity and GM profile alterations. Specifically, a knockout (KO) mouse model for the negative regulator of the mitochondrial complex I, methylation-controlled J (MCJ) protein, showed enhanced respiration rates and ATP synthesis without any increase in ROS production. NASH induction was obtained by feeding wild-type (WT) and MCJ KO mice on a choline-deficient, L-amino acid-defined, high-fat diet (CDA-HFD) for 6 weeks. MCJ KO mice showed improved intestinal barrier integrity and histopathological disease scores, as well as a peculiar microbiome profile, which was characterized by increased relative abundance of *Dorea* and *Oscillospira* and reduced proportions of *AF12*, *Allboaculum*, and [*Ruminococcus*]. When transferred to germ-free (GF) mice through cecal microbiota transplantation, the microbiome signature of MCJ KO mice exerted beneficial effects and delayed NASH progression mice by enhancing fatty acid oxidation, SCFA, nicotinamide adenine dinucleotide (NAD+) metabolism, and sirtuin activity. In particular, *Dorea* was identified as one of the key modulators of this protective phenotype; it was consistently found to be reduced in lean, but not obese, NAFLD individuals, highlighting the need to consider BMI when proposing therapeutic approaches based on microbiome modulation.

Regarding disease progression, Huang et al. investigated the GM composition in NAFLD and NASH patients, describing the changes that may be associated with clinical evolution [81]. A decrease in α-diversity was reported from healthy controls to NASH patients, with significant differences in taxonomic composition among the three groups. At the phylum level, both the NAFLD and NASH groups showed enrichment of Bacteroidetes and Fusobacteria, as well as decreased Firmicutes to Proteobacteria ratios; however, Firmicutes were higher and Bacteroidetes lower in NASH compared to NAFLD patients. At the genus level, higher levels of *Megamonas* and *Fusobacterium* were detected in the NASH group compared to the NAFLD group. When testing the predictive potential of the microbiome, the relative abundance of *Prevotella_9* was found to discriminate NAFLD patients from healthy controls, while the combination of *Megamonas* and *Fusobacterium* specifically identified the NASH group, suggesting that their elevated levels could be associated with disease progression and used as possible diagnostic tools. Notably, expected dissimilarities in GM composition were found between Asian and European NAFLD cohorts, likely due to differences in dietary habits and living environments, confirming that caution should be taken when generalizing microbiome results from different geographic areas. Diets differ significantly among countries due to differences in their development levels, cultural practices, and types of diets (omnivorous versus vegetarian). Nutritional components, such as fats, are reported to be associated with increases in *Bacteroidetes* and *Actinobacteria* species and decrease in Firmicutes and Proteobacteria phyla [82]. Notably, the host location showed the strongest associations with GM variations, exceeding the effect of metabolic diseases, including T2DM, metabolic syndrome, obesity, and fatty liver, confirming that caution should be taken when generalizing microbiome results from different geographic areas [83]. Moreover, whether the geographical effect is driven by host-specific factors (e.g., development during infancy, antibiotic treatments) or introduced by other ecological processes (e.g., infections by pathogens and host interactions with the environmental microbiome) requires further investigation [84].

Astbury and colleagues reported a progressive decrease in α-diversity from healthy controls to NASH without cirrhosis to NASH with cirrhosis [85]. The most representative OTU of NASH was *Collinsella*, showing an increased abundance from controls to NASH without cirrhosis to NASH with cirrhosis; it was positively associated with fasting triglycerides and total cholesterol and negatively associated with HDL, possibly influencing host lipid metabolism. Furthermore, the decrease in SCFA-producing genera of *Ruminococcaceae* in both NASH-related groups has been suggested to contribute to exacerbate liver inflammation, leading to disease progression. Finally, an interesting study by Smirnova et al. [57] reported a strong association between bile acid-producing bacteria and the degree of fibrosis in NAFLD and NASH cohorts. BAs of microbial origin increased with disease progression and were linked to the altered expression of BSH, Bai, and hydroxysteroid dehydrogenases (hdhA) genes, which are required for DCA and downstream metabolite synthesis, explaining how the GM might be responsible for changes in the fecal BA profile. In particular, bacteria harboring genes for DCA metabolism, such as members of the phylum Bacteroidetes and the family *Lachnospiraceae*, increased in parallel with disease severity. On the contrary, beneficial microbes sensitive to the antibacterial effects of DCA (e.g., *Ruminococcaceae*) decreased, emphasizing the profound effect of bile acid composition on microbiome remodeling. Mechanistically, DCA increases collagen synthesis by hepatic stellate cells via binding to TGR5 receptors [86] and leads to perturbation of the mitochondrial-mediated cell death pathway [87].

In conclusion, an entangled crosstalk between host characteristics and the gut and liver microbiome influences NAFLD progression, opening the way for the discovery of biomarkers for the different stages of the disease and therapeutic strategies based on microbiome modulation.

### 3.2. Promoting Healthy Lifestyles and Modulating Gut Microbiome Composition for the Treatment of NAFLD: Clinical and Preclinical Studies

Some evidence has shown that weight loss (at least 3–5%) generally reduces hepatic steatosis, with greater weight loss (up to 10%) also improving necroinflammation [88]. Although achievable in the trial setting, weight loss can be challenging in clinical practice and requires focused intervention in specialist clinics with the aim of long-term maintenance. Notably, plant-based diets are associated with lower NAFLD risk and more favorable liver function serum profiles [89]. In the absence of approved pharmacological therapies for NAFLD treatment, prior to the accelerated approval of Resmetirom last year [90,91], hypocaloric diets alone or in combination with incremental physical activity programs were effective methods for achieving consistent weight loss and improving hepatic steatosis and insulin sensitivity, and were therefore recommended by European and American guidelines as first-line interventions for the prevention and treatment of NAFLD [92,93].

A randomized clinical trial (RCT) in overweight/obese NASH patients reported the beneficial effect of weight reduction (>7%) on lobular inflammation, ballooning, and NAFLD activity score (NAS) [94]. After 48 weeks, a significant weight reduction (9.3%) and improvement in NAS were observed in the lifestyle intervention group, focused on diet, exercise, and behavior modification, compared to the control group. Similarly, Cheng et al. conducted an RCT in prediabetic patients with NAFLD comparing three different experimental groups: progressive aerobic exercise training (60–75% VO2max intensity for 2–3 times/week in 30–60 min/sessions), diet intervention (38% carbohydrate and 12 g/day fiber), and the combination of the two, for a period of 6 to 8 months [95]. Hepatic fat content (HFC), assessed by magnetic resonance (1H MNR), decreased in all intervention groups compared to the control group, with the highest decrease in the diet plus exercise group. No remission or progression of prediabetes to diabetes was observed in the intervention groups as measured by glycated hemoglobin (HbA1c). Interestingly, 16S rRNA sequencing analysis showed a deterioration in microbial composition with decreased α-diversity in the non-intervention group only, confirming a strong association between disease progression and the health status of the GM [96]. In addition, enrichment of amplicon sequence variants (ASVs) belonging to *Bacteroides* and *Ruminococcus* was observed in all three intervention groups, which also showed, at the metagenomic analysis, alterations in metabolic pathways, such as ‘carbohydrate metabolism’, ‘energy metabolism’, and ‘lipid metabolism’. Furthermore, the individual baseline gut microbial network predicted the response to the exercise intervention, but not to the diet-based interventions. Another study by Calabrese et al. identified an increased relative abundance of six genera (*Ruminococcus*, *Oscillospiraceae-UCG002*, *Oscillospiraceae-UCG005*, *Dialister*, *Alistipes*, and *Eubacterium eligens*) in the Mediterranean diet (MD) plus aerobic physical activity group compared to either the single treatments or the control diet [97]. The presence of fiber in the MD could positively influence the intestinal barrier, as demonstrated by Krawzyck et al., who reported a reduction in serum ZO-1, a surrogate marker for the assessment of “leaky” gut, in NAFLD patients after a 6-month dietary intervention with increased fiber consumption [98]. At the end of treatment, a positive correlation was reported between serum ZO-1 levels and biochemical parameters of hepatic damage (ALT, AST), serum triglycerides, and fatty liver status, suggesting an overall beneficial effect of increased dietary fiber intake. A weakness of this study is the lack of a control group, which limits the biological relevance of the findings. Adherence to the MD, as evaluated by the MedDietScore, was negatively correlated with serum levels of liver damage parameters, severity of liver steatosis, and insulin levels. Patients with NASH were less adherent to the MD than those with NAFLD, indicating that adherence to the MD was also inversely correlated with liver disease severity [99]. A meta-analysis showed that prebiotics, such as psyllium, *Ocimum basilicum*, and inulin, led to improvements in anthropometric and biochemical parameters, suggesting the need for prospective controlled studies to better define fiber type, dosage, and duration interval for optimal results [100].

Regarding the influence of diet composition, a Chinese study reported the effect of a freshwater fish-based (F) diet compared to an isocaloric freshwater fish-/red meat-based (F/M) diet in NAFLD individuals [101]. At the end of the intervention (84 days), the absolute decrease in hepatic steatosis and the relative reduction in liver fat content were greater in the F group than in the F/M group, in line with the improvement in several biochemical parameters. Notably, the F group also showed an enrichment of *Faecalibacterium* (a SCFA producer candidate as a next-generation probiotic), fecal SCFAs, and fecal unconjugated BAs, as well as a parallel depletion of *Prevotella_9* and conjugated BAs. These changes correlated with the liver profile, suggesting that gut microbiome-induced metabolic alterations might be responsible for the clinical improvement in NAFLD patients on a fish-based diet rich in ω-3 polyunsaturated fatty acids (n-3 PUFAs). Indeed, preclinical studies in NAFLD mouse models fed a high-fat, methionine choline-deficient (MCD) diet demonstrated the beneficial effect of n-3 PUFAs in decreasing serum triacylglycerol and cholesterol concentrations and hepatic triacylglycerol content, suggesting promising nutraceutical effects for the prevention and treatment of NAFLD [102]. However, a meta-analysis reported a significant benefit of PUFA supplementation in reducing liver fat, but not transaminase, levels, highlighting a huge variation between studies, with the optimal dose yet to be identified [103].

Interestingly, a preclinical study reported the beneficial effect of 16:8 time-restricted feeding (TRF) in a high-fat-diet NAFLD mouse model [104]. After 6 weeks, hepatic steatosis improved in the TRF group, with decreased expression of aldehyde oxidase 1 (AOX1), a key enzyme involved in nicotinamide (NAM) catabolism, which mediates de novo lipogenesis and fatty acid uptake through the production of pro-steatotic metabolites. Notably, FMT from TRF-treated mice to HFD-fed mice mimicked the effect of the TRF regimen on NAM catabolism, reducing hepatic fat content and improving lipid metabolism, demonstrating that GM mediates the metabolic modulatory effect of TRF on NAM metabolism and providing novel insights for the prevention and treatment of NAFLD. In line with this, a TRF regimen was also shown to alleviate obesity and NASH progression in mice by restoring the rhythmicity of genera such as *Lactobacillus*, *Mucispirillum*, *Acetatifactor*, and *Lachnoclostridium* [105]. Similarly, every-other-day fasting (EODF) modulated the GM composition and increased the levels of the fermentation end-products acetate and lactate [106], turning out to be an interesting approach for the treatment of metabolic disorders such as obesity and NASH. Regarding intermittent fasting, Lin and coworkers performed a preclinical study in a mouse model of NASH obtained by feeding animals a high-fat, high-cholesterol (HFHC) diet for 16 weeks with an intervention group treated with EODF for another 10 weeks [107]. EODF-treated animals showed reduced body weight, insulin resistance, hepatic steatosis, and components of the NAS score such as ballooning and lobular inflammation. A reshaping of the GM was also observed in EODF-treated mice together with a reduction in serum total BAs, suggesting that intermittent fasting may have a protective effect against NASH by regulating microbial metabolism of BAs and promoting their intestinal excretion. In addition, EODF decreased the relative abundance of Cyanobacteria and *Lactobacillus* while increasing that of *Lachnospiraceae*, *Peptococcaceae*, *Peptococcus*, *Butyricicoccus*, and *Blautia*. Intermittent fasting may act on NAFLD pathogenesis through several mechanisms, such as switching towards β-oxidation for energy production, reducing inflammation, and modulating the GM [108].

The modulation of the gut microbiome by a probiotic mixture (Lactobacillus acidophilus, Lactobacillus rhamnosus, Pediococcus pentosaceus, Bifidobacterium lactis, Bifidobacterium breve, Lactobacillus paracasei) was investigated in a double-blind RCT study in obese NAFLD individuals [109]. All the microbial strains except for L. paracasei increased in the probiotic group after a 12-week supplementation, while P. pentosauceus was only increased in the placebo group. The authors showed a decrease in triglyceride levels compared to the placebo arm, but no clear results were obtained in terms of changes in liver fat composition, highlighting the need for further investigations before proposing this supplementation strategy in NAFLD patients. Similarly, a double-blind phase 2 trial evaluating the effect of 1-year supplementation with a synbiotic formulation (fructo-oligosaccharides plus Bifidobacterium animalis subspecies lactis BB-12) reported higher proportions of Bifidobacterium and *Faecalibacterium* spp. and reduced proportions of Oscillibacter and *Alistipes* spp., but no associations with changes in liver fat content or markers of liver fibrosis [110]. According to a recent meta-analysis, synbiotics can ameliorate liver function, anthropometric parameters, and inflammatory markers, but does not lead to significant changes in liver fibrosis and steatosis or body composition, indicating that further adjustments in supplementation strategies or combinations with lifestyle interventions are needed [111]. Finally, preclinical data showed a beneficial effect of A. muciniphila supplementation in an obese MAFLD mouse model obtained by feeding animals an HFHC diet [112]. Oral gavage with this potential next-generation probiotic decreased body weight, liver triglycerides, inflammatory cytokines, and liver injury markers and, notably, had a long-lasting effect. From a mechanistic point of view, mice treated with A. muciniphila activated energy expenditure by increasing lipid oxidation and mitochondrial copy number, confirming the anti-obesogenic effect observed in preclinical models [113] and providing support for the use of A. muciniphila preparations to target NAFLD and associated metabolic disorders. A. muciniphila has recently been approved as a postbiotic for metabolic health, namely weight management and glycemic control [114,115]. Probiotics hold promise for the management of NAFLD patients, but, since no consensus about their efficacy is reported in the literature [116], further preclinical studies and clinical trials are highly recommended before translating the preclinical findings into clinical practice.

FMT represents a promising approach for the treatment of dysbiosis-related diseases [117], including NAFLD, as demonstrated by preclinical studies [118,119]. An RCT study reported the efficacy and safety of colonoscopy-based FMT from healthy donors to NAFLD recipients [120]. Metagenomics analysis revealed that FMT improved the diversity and composition of the GM in NAFLD patients, increasing the relative abundance of Bacteroidetes and the Bacteroidetes-to-Firmicutes (B/F) ratio. Interestingly, when the FMT group was stratified by BMI, significant post-treatment differences in the average fat attenuation index and GM composition were observed between obese and lean NAFLD patients, with the latter showing greater clinical efficacy of FMT. In line with this, another double-blind RCT study reported that allogenic FMT from lean vegan donors to NAFLD individuals led to beneficial changes in gut microbiome composition, blood metabolites, and markers of steatohepatitis [121]. In particular, allogenic FMT increased the relative abundance of *Ruminococcus*, *Eubacterium hallii*, *Faecalibacterium*, and *Prevotella copri*, while autologous FMT resulted in minor changes except for increased proportions of *Lachnospiraceae*. Furthermore, plasma levels of detrimental metabolites, such as phenyllactic acid—a harmful microbial product of aromatic amino acid metabolism—and desaminotyrosine, were increased following autologous FMT only. FMT has been demonstrated as an effective therapeutic approach for recurrent *Clostridium difficile* infection (CDI), and its adoption in the clinical practice has circumvented the standard regulatory processes (e.g., RCT studies) that enable the collection of important safety data [122]. Because FMT meets the legal definitions of both a biologic product and a drug, this practice cannot be used for any indication other than unresponsive/recurrent CDI without an ‘investigational new drug’ [123]. Indeed, uses of FMT that extend beyond pathogen transmission (e.g., obesity, diabetes, cancer) raises several safety concerns; in addition, the absence of consensus guidelines on practice standards may affect the short and long-term risks of this therapeutic approach. Although exciting results have been achieved in understanding GM and its modulation, more data and regulatory actions are needed to increase knowledge about the efficacy and safety of FMT in human disease.

In summary, clinical and preclinical studies have demonstrated the positive impact of gut microbiome modulation on the improvement of NAFLD clinical parameters, highlighting a wide spectrum of possible interventions (Figure 2). However, large inter-individual variability has been observed in NAFLD patients, e.g., in baseline microbiome profiles, BMI, etc., which should be taken into account in the design of personalized precision strategies with improved efficacy.

### 3.3. Gut Microbiome Restoration in the Prevention of NAFLD Progression to HCC: Preclinical Studies

Given the high incidence of NAFLD worldwide, NAFLD and NASH are becoming the most frequent etiologic factors for HCC; in fact, NAFLD patients with high-grade fibrosis have a sevenfold higher risk of tumor development in comparison to those without liver disease. NASH-related HCC is increasingly diagnosed in the clinical setting and is the most rapidly growing HCC indication for liver transplantation in the United States [124]. Since no established therapeutic approaches are available to inhibit NAFLD progression in clinical practice, here, we will review the most recent studies investigating the modulation of the GM to prevent/slow down the transition from NAFLD/NASH to HCC in the preclinical setting. Indeed, despite the recent approval of Resmetirom for MAFLD treatment [90,91], information regarding HCC prevention is not yet available.

Li and coworkers assessed the antitumor potential of *A. muciniphila* in a NASH-HCC mouse model called STAM, resembling human pathology, which was obtained following streptozotic (STZ) injection immediately after birth and feeding with an HFD [125]. The study reported a reduced relative abundance of *A. muciniphila* in NAFLD-HCC patients and STAM mice and demonstrated that oral supplementation with this taxon improved clinical parameters of NASH (steatosis, ballooning, NAS) and reduced HCC progression. *A. muciniphila* supplementation also reshaped the inflammatory microenvironment, decreasing IL-6 levels and favoring NKT cell infiltration in the liver, which mediated the antitumor effect. Notably, the modulation of the immune cell compartment and the reduction in the pro-inflammatory chemokine IL-17 have been identified as mechanisms of action of a probiotic mixture tested in HCC mice, leading to the improvement of gut dysbiosis and, in particular, to the enrichment of *A. muciniphila* [126]. The study by Song et al. reported the beneficial effects of *Bifidobacterium pseudolongum* administration in two mouse models of NAFLD-HCC (DEN + HFHC diet or + a choline-deficient/HFD). *B. pseudolongum* suppressed tumor formation in both animal models and restored a healthy GM composition, also improving gut barrier function [127]. Acetate was found to be the critical metabolite that mediated most of the antitumor functions, as demonstrated by in vitro and in vivo studies. Mechanistically, acetate produced by *B. pseudolongum* reached the liver via the portal vein, where it bound to GPR43 (G-protein-coupled receptor 43), whose activation blocked the IL-6/JAK1/STAT3 signaling pathway, thereby blocking pro-inflammatory cascades and preventing NAFLD-HCC progression. The same authors proved the antitumor potential of another probiotic bacterium, *Lactobacillus acidophilus*, in the same NAFLD-HCC animal models cited above, as well as in orthotopic allografts and germ-free tumorigenesis mice [128]. Valeric acid was identified as the most abundant and prominent metabolite mediating *L. acidophilus* anticancer effects in 2D and 3D in vitro models and in HFHC-fed DEN-treated mice. Specifically, valeric acid was found to be upregulated in the mouse liver, where it suppressed the Rho-GTPase pathway by binding to GPR41/43 and inducing cell cycle arrest. An outstanding study by Zhang and coworkers demonstrated that the anti-cholesterol drug atovarstatin was able to prevent HCC onset in mice fed an HFHC diet by modulating the GM [129]. Dietary cholesterol caused spontaneous NAFLD–HCC formation that was associated with an imbalance in the GM: *Mucispirillum*, *Desulfovibrio*, *Anaerotruncus*, and *Desulfovibrionaceae* were sequentially increased with the different stages of the disease, while *Bifidobacterium* and *Bacteroides* were depleted and inversely correlated with cholesterol levels in humans. FMT from HFHC-fed mice into GF mice triggered hepatic lipid accumulation, cell proliferation, and inflammation in recipient animals, proving that gut dysbiosis was the cause, not the effect, of the onset of NAFLD-HCC following a high-cholesterol diet. Notably, BA biosynthesis was a key pathway enriched in HFHC-fed mice, which showed higher serum levels of primary BA metabolites (e.g., taurocholic acid—TCA) and lower levels of the microbial tryptophan metabolite, 3-indolepropionic acid (IPA). Mechanistically, TCA aggravated cholesterol-induced triglyceride accumulation in normal immortalized hepatocytes, whereas IPA inhibited cholesterol-induced lipid accumulation and cell proliferation. Thereby, the atorvastatin-mediated decrease of cholesterol, which is a major lipotoxic molecule that induces ROS and proinflammatory cytokines, together with the restoration of GM diversity, could ameliorate metabolic functions of liver cells through the modulation of GM-derived metabolites, preventing NAFLD progression. Because FMT from atorvastatin-treated HFHC-fed mice did not promote hepatic cell proliferation in recipient GF mice and atorvastatin treatment prevented steatohepatitis and fibrosis and HCC development in HFHC-fed mice, statin treatment represents a promising and cost-effective strategy to prevent/delay disease progression in hypercholesterolemic NAFLD patients.

These recent preclinical findings confirm the active role of GM dysbiosis in NAFLD progression to HCC (Figure 3) and pave the way for the administration of anti-cholesterol drugs or probiotics in NAFLD patients with or without hypercholesterolemia from a preventive perspective.

## 4. Systemic Treatments for Advanced HCCs

Because of the late diagnosis of advanced cases, high recurrence rates following surgical resection and progression to locoregional therapy, approximately 50% of HCC patients ultimately receive systemic therapy [130]. ICIs have dramatically changed the treatment landscape for advanced HCC, with two trials demonstrating superior overall survival benefits over sorafenib in first-line settings [131,132]. Immunotherapy has greatly prolonged the life expectancy in responder patients, showing a median survival of 19.6 months in the real world [133]. Nevertheless, the huge genetic heterogeneity of HCC and the lack of reliable biomarkers for stratification are at the basis of immune escape in certain subgroups of patients, such as those presenting with β-catenin mutations [134].

The GM plays an important role in shaping the liver microenvironment and suppressing antitumor immunity, favoring HCC progression in several preclinical models [135]. Below, we summarize the most recent articles reporting the alterations of the GM in HCC patients undergoing immunotherapy, as well as the last research efforts focused on the modulation of the GM as a new therapeutic strategy for combined and tailored treatments.

### Influence of the Gut Microbiome and Its Modulation on Response to Immunotherapy in HCC

Over the past decade, immunotherapy has achieved great success against several metastatic solid malignancies and has also gained approval for HCC treatment in the first line [136]. Despite its impressive efficacy, immunotherapy induces sustained clinical responses in a minority of cancer patients. An outstanding study demonstrated that an altered GM, as well as antibiotic treatment, can jeopardize ICI response in patients with epithelial tumors [137]. As proof of principle, the authors showed that FMT from responder patients into germ-free mice improved the antitumor effects of anti-PD-1 monoclonal antibodies. Moreover, *A. muciniphila* was significantly associated with a favorable clinical outcome and was enriched in patients with longer progression-free survival (PFS); its supplementation restored anti-PD-1 blockade in mice subjected to FMT from non-responder patients. Mechanistically, *A. muciniphila* induced the secretion of IL-12 by dendritic cells, increasing the recruitment of CCR9+CXCR3+CD4+ T cells into mouse tumors. Interestingly, an observational study in patients with advanced digestive tract cancers (HCC, colorectal, and gastric cancers) undergoing combined immunotherapy–antiangiogenic treatment reported a direct correlation between probiotic supplementation and a higher ORR or prolonged PFS, supporting the hypothesis that the gut microbiome may play a key role in the response to immunotherapy [138]. Because the study did not specify the composition of probiotics, to determine which probiotic strains contribute synergistically to the ICIs response, specific treatment information needs further investigations. In the context of co-medications, a retrospective study in advanced HCC patients receiving ICIs (nivolumab, pembrolizumab, or ipilimumab) showed that concurrent antibiotic use was associated with higher cancer-related and all-cause mortality [139]. In contrast, a large international cohort study (Europe, North America and Asia) reported a positive effect of early antibiotic exposure (−30 and +30 days from ICIs) on PFS in HCC patients receiving ICIs after one first-line treatment [140]. In patients receiving anti-PD-1/PD-L1 monotherapy, co-medication with antibiotics was also associated with a higher disease control rate. Because of contrasting findings showing detrimental effects of antibiotics not only in the ICI group, but also in the TKI and placebo groups [141], the evaluation of the microbiological determinants of response to ICIs in HCC warrants further investigation.

Regarding the influence of the GM on ICI response in patients with hepatobiliary cancers (unresectable HCC and advanced biliary tract cancers that have progressed from first-line chemotherapy), Mao et al. reported the enrichment of several taxa in responders [142]. Among these, *Lachnospiraceae bacterium-GAM79* and *Alistipes sp Marseille-P5997* were significantly enriched in the group that achieved clinical benefit in terms of prolonged OS and PFS. Although the results reported in this pioneering study are interesting, the heterogeneity of the patient cohort in terms of cancer types will probably make it difficult to compare them with other studies.

With specific regard to HCC, Lee et al. highlighted the association of fecal microbiome and BAs with ICI treatment outcomes [143]. In particular, *Prevotella 9* was depleted in patients with objective response (OR), whereas *Lachnoclostridium*, *Lachnospiraceae*, and *Veillonella* were enriched. Ursodeoxycholic acid and ursocholic acid were also enriched in the feces of patients with OR and positively correlated with *Lachnoclostridium*. The strength of this study lies in the presence of a validation cohort in which the coexistence of *Lachnoclostridium* enrichment and *Prevotella 9* depletion predicted a favorable outcome in terms of both OS and PFS, highlighting the potential of the GM and related metabolites as prognostic biomarkers in HCC patients undergoing immunotherapy. In line with this, Zheng et al. reported the enrichment of a number of potentially beneficial taxa in responders, including *Lachnospiraceae*, *Ruminococcaceae*, *A. muciniphila*, *Lactobacillus species*, *Bifidobacterium dentium*, and *Streptococcus thermophilus* [144]. The limitation of this study was the small sample size, including only eight sorafenib-progressor patients treated with anti-PD-1 for metagenomics analysis. Similarly, Ponziani et al. reported interesting findings regarding the dynamic changes of the GM and intestinal inflammation parameters during ICI treatment cycles [145]. Specifically, they reported decreased basal levels of fecal calprotectin and serum PD-L1 in patients with disease control, together with increased proportions of *Akkermansia* and decreased proportions of *Enterobacteriaceae*. An opposite temporal trend, namely an increase in calprotectin and a decrease in the ratio of *Akkermansia* to *Enterobacteriaceae* (A/E), was detected at the time of patient discontinuation. Despite the small sample size (11 patients treated with Tremelimumab and/or Durvalumab), this study highlighted the promise of the GM, inflammation parameters, and serum metabolites as possible prognostic biomarkers.

Interestingly, a multi-kingdom microbiome characterization, including bacteria and fungi, as well as metabolites, showed that bacteria and metabolites, but not fungi, significantly differed between patients with durable (>6–8 weeks) and non-durable clinical benefit [146]. The model, including 18 bacterial taxa, predicted ICI clinical efficacy, whereas two bacterial species (*Actinomyces* sp *ICM47* and *Senegalimassilia anaerobia*) and one metabolite (galanthaminone) showed prognostic value for survival in ICI-treated patients. In the same context, a serum metabolite classifier demonstrated better predictive potential in discriminating HCC patients who potentially benefited from immunotherapy than a gut microbiome-based classifier [147]. This result may be explained by the fact that microbial metabolites directly modulate key immunological processes that positively or negatively influence liver cancer’s susceptibility to immunotherapy [148].

In the preclinical setting, the anticancer effect of *A. muciniphila* in combination with an anti-PD-1 monoclonal antibody was assessed in a xenograft mouse model [149]. The combined treatment showed a higher antitumor effect, mainly associated with an increase in INF-γ-producing CD8+ T cells and a decrease in PD-L1 expression in the tumor, together with an increase in serum BA metabolites. However, it should be noted that the xenograft model is likely not the best experimental tool due to the absence of the gut–liver axis and the proper tumor microenvironment, making it difficult to extrapolate the results regarding the mechanisms that influence the host immune response following *Akkermansia* supplementation.

Analysis of the GM is expected to provide a huge amount of data on its association with treatment outcomes in the foreseeable future. Once its role is clarified, further efforts and preclinical investigations are expected to develop and test microbiome-based therapeutic interventions, e.g., based on probiotics or postbiotics, to boost the efficacy of, or overcome the resistance to, immunotherapy-based approaches for advanced HCC.

## 5. Conclusions and Future Perspectives

In the first part of this review, we discussed the impact of a variety of host metadata, including diet composition, lifestyle, medications (e.g., antibiotics and statins), and comorbidities (e.g., diabetes, obesity), on the GM in patients with NAFLD, chronic liver diseases, and HCC. A future challenge will be the reproducibility of studies, which may affect the discovery of diagnostic and prognostic signatures based on omics profiling of GM and derived metabolites. Longitudinal sampling [150] and adequate control groups are needed to compare different studies and to strengthen associations between microbial signatures and disease progression, thereby enhancing hypothesis-driven microbiome research in chronic liver diseases.

In the second part, we reported clinical studies describing the presence of associations between specific intestinal bacterial species and/or microbial-derived metabolites in ICIs responder HCCs. In this context, studies in larger patient cohorts are necessary to better define the influence of GM on immunotherapy response, whereas preclinical studies will help provide mechanistic insights, including how the GM reshapes the tumor microenvironment, boosting the immune response and jeopardizing immune evasion of HCC cells. In this regard, some evidence has suggested the involvement of *A. muciniphila* in several contexts such as NAFLD/NASH progression, HCC development, and immunotherapy response, and has shown the antitumor effects of its restoration in HCC animal models (Figure 4). Further functional studies are needed to better understand the bidirectional and dynamic interactions between human liver cells, microbes, and their metabolites, which directly and/or indirectly impact the metabolism and energy status of hepatocytes, as well as the inflammatory milieu through the activation of stroma and immune cell subtypes.

All these aspects will hopefully lead to the design of focused clinical trials to test microbiome-tailored strategies, such as prebiotics, (traditional or next-generation) probiotics, synbiotics, postbiotics, and other strategies (e.g., FMT or ad hoc synthetic consortia) aimed at modulating and restoring the GM to treat CLD and prevent its progression. However, it is important to note that many challenges remain in the development of precision GM modulation strategies, such as GM and host variability, potential risks of live microbial supplementation, and the difficulty of standardizing GM-based interventions in a clinical setting. In particular, microbiome-associated confounding factors, such as age, gender, ethnicity, lifestyle, geographic location, and/or the exposome in general [38], should be taken into account, as they may lead to differential treatment responses. Furthermore, it has been shown that, if not tailored, probiotic interventions may not only be ineffective, but also not entirely risk-free [151]. Similarly, safety concerns (including the potential risk of transmission of enteric multidrug-resistant organisms) have been highlighted for FMT [152], driving research towards the design of artificial (and controlled) consortia or even postbiotics to directly deliver the molecular actors of interest in the specific context. Despite such challenges, the evidence describes how the effectiveness of probiotic strains might not rely on colonizing the gastrointestinal tract, but rather reside in their capacity of sharing genes and secreting metabolites, restoring the growth of commensal bacteria [153]. In this regard, potential risks of probiotic supplementation should also be considered, as they can produce deleterious metabolites and transfer antibiotic resistance genes in commensals or potential pathogens, leading to dysbiosis or overgrowth of certain taxa. A further aspect to take into consideration when considering GM-based interventions in the clinical setting is related to the basal variability in GM across individuals and distinct geographic regions. The responses to the same probiotic may considerably differ due to variabilities such as host genetics, diet, and endogenous GM composition, suggesting that the basal GM of each person influences the response to different probiotic strains [154]. In patients with CLD, this aspect will be worsened even further by disease-associated dysbiosis, complicating the standardization of GM-based interventions.

## Figures and Tables

**Figure 2 cells-14-00084-f002:**
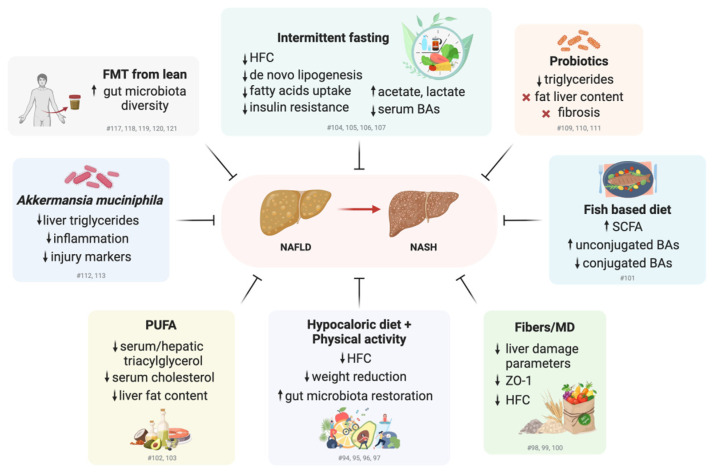
Healthy lifestyles and gut microbiome modulation concur to reduce NAFLD progression to NASH by reducing steatosis and improving markers of hepatic damage. HFC: hepatic fat content; BAs: bile acids; SCFA: short-chain fatty acids; MD: Mediterranean diet; ZO-1: zonulin-1; PUFA: polyunsaturated fatty acids; FMT: fecal microbiota transplantation. Numbers preceded by “#” indicate the reference article.

**Figure 3 cells-14-00084-f003:**
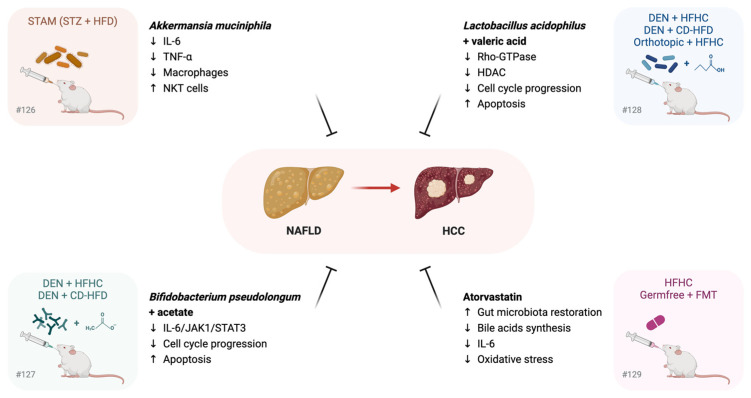
Modulating gut microbiome by probiotics and postbiotics or Atorvastatin reduces NAFLD progression to HCC in animal models of NAFLD/NASH-HCC. STZ: streptozotocin; HFD: high-fat diet; IL-6: interleukin 6; TNF-α: tumor necrosis factor α; NKT cells: natural killer T cells; HDAC: histone deacetylase; DEN: diethyl nitrosamine; HFHC: high-fat high-cholesterol; CD-HFD: choline-deficient high fat diet; JAK1: Janus kinase 1; STAT3: signal transducer and activator of transcription 3; FMT: fecal microbiota transplantation. Numbers preceded by “#” indicate the reference article.

**Figure 4 cells-14-00084-f004:**
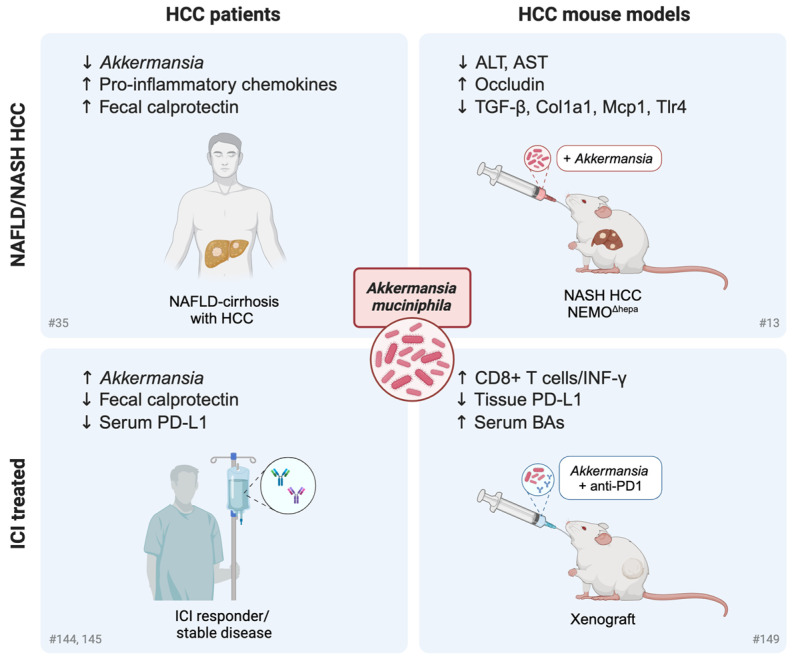
*Akkermansia muciniphila* reduces HCC progression and improves ICI treatment response in HCC patients and mouse models. ALT: alanine transaminase; AST: aspartate transferase; TGF- β: transforming growth factor β; Col1a1: Collagen type I alpha 1 chain; Mcp1: Monocyte chemoattractant protein-1; Tlr4: Toll-like receptor 4; NEMO^Δhepa^: steatohepatitis-HCC mouse model with NEMO gene deletion; ICI: immune checkpoint inhibitors; PD-L1: programmed cell death ligand 1; INF-γ: interferon γ; BAs: bile acids; PD1: programmed cell death protein 1. Numbers preceded by “#” indicate the reference article.

**Table 1 cells-14-00084-t001:** Dysregulation of gut microbiome in HCC patients.

Comparison	HCC Etiology	Dysregulated Microbiome	Methodology	Reference
eHCC vs. cirrhosis	HBV infection	Actinobacteria phylum ↑, *Gemmiger* ↑, *Parabacteroides* ↑, *Paraprevotella* ↑	16S rRNA amplicon sequencing	[33]
NAFLD-HCC vs. NAFLD-cirrhosis	NAFLD	*Enterococcus* ↑, *Streptococcus* ↑, *Akkermansia* ↓, *Bifidobacterium* ↓	16S rRNA amplicon sequencing	[35]
NAFLD-HCC vs. NAFLD-cirrhosis	NAFLD	*Bacteroides caecimuris* ↑, *Veillonella parvula* ↑, *Clostridium bolteae* ↑, *Ruminococcus gnavus* ↑, *Enterobacteriaceae* ↑	Shotgun metagenomic sequencing	[36]
HCC vs. NAFLD	AFLD, NAFLD, HBV/HCV infection, PBC, PSC, AIH	*Blautia* ↓, *Agathobacter* ↓, *Ruminococcaceae* ↑, *Bacteroidaceae* ↑	16S rRNA sequencing	[37]

eHCC: early HCC, HBV: hepatitis B virus, HCV: hepatitis C virus, NAFLD: non-alcoholic fatty liver disease, AFLD: alcohol-associated fatty liver disease, PBC: primary biliary cholangitis, PSC: primary sclerosing cholangitis, AIH: autoimmune hepatitis.

## Data Availability

Not applicable.

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
