# Peer review of "Gut Microbiome Modulation in Hepatocellular Carcinoma: Preventive Role in NAFLD/NASH Progression and Potential Applications in Immunotherapy-Based Strategies"

_cells, 2025, doi:10.3390/cells14020084_

Round 1

Reviewer 1 Report

Comments and Suggestions for Authors

The manuscript, “Gut microbiome modulation in hepatocellular carcinoma: preventive role in NAFLD/NASH progression and potential applications in immunotherapy-based strategies”, addresses a highly relevant and timely topic. It focuses on the modulation of the gut microbiome in the context of hepatocellular carcinoma (HCC), emphasizing its role in preventing NAFLD/NASH progression and its potential applications in immunotherapy-based strategies.

I recommend major revisions before considering the manuscript for publication in a highly reputed journal such as cells.

1.     Authors have mentioned the gut-liver axis and microbial metabolites in abstract. Specify the key microbial metabolites or pathways involved in the abstract.

2.     Elaborate the role of GM in influencing immunotherapy response with example or mechanism to strengthen the point in the abstract.

3.     Include specific statistics for GM diversity changes and response rates to interventions in the abstract to add more weight to the claims.

4.     Line 36- "six most common cancer" should be corrected to "sixth most common cancer."

5.     Line 41- Mention full forms of HBV and HCV. Full forms of abbreviations should be given on first mention.

6.     Line 75- The role of GM derived metabolites is mentioned. Include one or two examples of the metabolites to enhance understanding.

7.     Line 81- The line mentions GM based treatments in the context of personalized medicine. The link between GM and precision medicine is not fully developed. Elaborate on how GM could guide tailored treatments.

8.     Line 105- Elaborate on the molecular mechanisms underlying the reduction in the microbe-impermeable inner mucus layer. Also add specific examples or references to studies that illustrate these mechanisms.

9.     Line 108- Changes in tight junctions and increase in permeability discussion could be expanded to include the specific proteins involved such as Claudin, Occludin, ZO-1 and how their expression or localization is altered in MASLD.

10.  Provide specific examples of microbial species or metabolites implicated in pro-tumorigenic effects beyond those already mentioned.

11.  Therapeutic targets of HCC are mentioned like GM restoration and TLR4 signaling. Discuss the challenges or limitations of targeting these pathways, such as potential off-target effects or variability in gut microbiome composition among individuals.

12.  Line 160- It would be helpful to discuss potential mechanisms or hypotheses explaining the increase in microbial diversity in early HCC compared to cirrhosis, as it contrasts with the typical reduction in diversity seen in other disease contexts.

13.  What is the mechanistic link between bacterial translocation and MT1B expression?

14.  Incorporate a table summarizing key microbial taxa and their roles in HCC progression, to enhance comprehension.

15.  Line 232-give references in support of the statement.

16.  Line 372: Add a period after “[55]”

17.  Figure 2: The name of the bacterium “Akkermansia munciniphila” in figure panel should be italicized.

18.  Use consistent terminology throughout ("gut microbiota" vs. "gut microbiome").

19.  Some mechanistic details of metabolites are mentioned such as the role of SCFAs in ROS production and apoptosis, but others are not explained in detail. Discuss how specific microbial metabolites or bile acids influence signaling pathways in depth.

20.  The role of isoallolithocholic acid in inducing immunosuppressive macrophages could be elaborated further to explain how this metabolite interacts with liver cells or pathways. Similarly, while discussing impact of DCA-producing bacteria on fibrosis progression, give clearer explanation of how these bacteria alter bile acid composition and affect liver function.

21.  How might dietary habits or environmental factors contribute to geographic differences in microbiota composition?

22.  Discuss the safety and ethical considerations of FMT, especially in clinical settings.

23.  How does atorvastatin mediated GM modulation and the prevention of NAFLD-HCC relates to cholesterol metabolism and GM-derived metabolites?

24.  Link preclinical findings to clinical implications, such as how probiotics or anti-cholesterol drugs could be integrated into current NAFLD management strategies.

25.  Figure 2: “Ipocaloric diet” please correct the spelling.

26.  Figure 3- Change “oxydative” to “oxidative”

27.  Line 662- Replace "no therapeutic approaches to inhibit NAFLD progression are available in the clinics" with "no established therapeutic approaches are available to inhibit NAFLD progression in clinical practice."

28.  Line 667, 670, 672, 673, 678: The name of bacterium “A. munciniphila” should be italicized.

29.  Line 679, 680, 684, 699: The name of bacterium “Bifidobacterium pseudolongum” should be italicized.

30.  Line 684, 692: terms “in vitro” and “in vivo’ should be italicized.

31.  Line 688, 691: The name of bacterium “Lactobacillus acidophilus” should be italicized.

32.  Line 698: The names of the bacteria “Mucispirillum, Desulfovibrio, Anaerotruncus” and the family “Desulfovibrionaceae” should be italicized.

33.  Figure 4: The name of the bacterium “Akkermansia munciniphila” in the figure caption should be italicized.

34.  Line 744- change “Authors” to “authors”

35.  Discuss the challenges presented by GM modulation such as, variability in GM composition across individuals and populations, potential risks of probiotic supplementation like dysbiosis or overgrowth of certain taxa, feasibility of standardizing GM based interventions in a clinical setting.

36.  Define "objective response rate (ORR)" for non-specialist readers.

Author Response

Reviewer 1

The manuscript, “Gut microbiome modulation in hepatocellular carcinoma: preventive role in NAFLD/NASH progression and potential applications in immunotherapy-based strategies”, addresses a highly relevant and timely topic. It focuses on the modulation of the gut microbiome in the context of hepatocellular carcinoma (HCC), emphasizing its role in preventing NAFLD/NASH progression and its potential applications in immunotherapy-based strategies.

I recommend major revisions before considering the manuscript for publication in a highly reputed journal such as cells.

  1. Authors have mentioned the gut-liver axis and microbial metabolites in abstract. Specify the key microbial metabolites or pathways involved in the abstract.

R: The key microbial metabolites investigated in this review, SCFAs and BAs, have been mentioned in the abstract.

  1. Elaborate the role of GM in influencing immunotherapy response with example or mechanism to strengthen the point in the abstract.

R: Two examples related to probiotic supplementation and anti-hypercholesterolemic drug treatment have been added in the abstract. We thank the reviewer for this suggestion, but we would prefer not to overload the abstract with mechanistic data regarding the role of GM in influencing the immunotherapy response, which have been deeply detailed in the text.

  1. Include specific statistics for GM diversity changes and response rates to interventions in the abstract to add more weight to the claims.

R: Thank you for the suggestion, but we would prefer not to overload the abstract with statistical data on GM diversity changes and response rates from the various studies mentioned in our review. In any case, only significant data from research articles have been discussed in the text, precisely to convey only significant claims to the reader.

  1. Line 36- "six most common cancer" should be corrected to "sixth most common cancer."

R: This correction has been performed.

  1. Line 41- Mention full forms of HBV and HCV. Full forms of abbreviations should be given on first mention.

R: Full name for HBV and HCV has been added.

  1. Line 75- The role of GM derived metabolites is mentioned. Include one or two examples of the metabolites to enhance understanding.

R: One example of GM derived metabolites (secondary bile acids) has been added together with a new reference reporting their role in shaping the gut microbiome composition and promoting hepatocarcinogenesis (new Ref. N.12). We thank the reviewer for this comment.

  1. Line 81- The line mentions GM based treatments in the context of personalized medicine. The link between GM and precision medicine is not fully developed. Elaborate on how GM could guide tailored treatments.

R: We added a sentence explaining how the GM could improve the efficacy of tailored treatments. We also added a new reference (new Ref. N.14). We thank the reviewer for this comment.

  1. Line 105- Elaborate on the molecular mechanisms underlying the reduction in the microbe-impermeable inner mucus layer. Also add specific examples or references to studies that illustrate these mechanisms.

R: We added two studies reporting the molecular mechanisms involved in the impairment of the inner mucus layer (new Ref. N.19, 20). We thank the reviewer for this comment allowing us to better detail the liver-to-gut axis.

  1. Line 108- Changes in tight junctions and increase in permeability discussion could be expanded to include the specific proteins involved such as Claudin, Occludin, ZO-1 and how their expression or localization is altered in MASLD.

R: We added another study reporting the altered expression of ZO-1 and PV-1 in a mouse model fed a high-fat diet, demonstrating the disruption of intestinal epithelial and gut vascular barriers as an early event in NAFLD-to-NASH progression (new Ref. N.22). We thank the reviewer for this comment allowing us to better describe the role of gut integrity in the pathogenesis of NAFLD.

  1. Provide specific examples of microbial species or metabolites implicated in pro-tumorigenic effects beyond those already mentioned.

R: We added a study reporting the alteration of microbial species in an HCC rat model showing the involvement of dysbiosis and gut inflammation in hepatocancerogenesis as well as the protective anticancer effect of probiotics administration (new Ref. N.26). Moreover, we added a very recent study describing the translocation of K. pneumonie into the liver of HCC patients leading to the activation of oncogenic pathways (new Ref. N.27). We thank the reviewer for this comment allowing us to integrate further information relative to this topic.

  1. Therapeutic targets of HCC are mentioned like GM restoration and TLR4 signaling. Discuss the challenges or limitations of targeting these pathways, such as potential off-target effects or variability in gut microbiome composition among individuals.

R: We critically discussed the possibility to use non-coding RNAs to target TLR4 signaling, highlighting pro and cons of this innovative strategy in HCC (new Ref. N.31, 32). We thank the reviewer for this comment.

  1. Line 160- It would be helpful to discuss potential mechanisms or hypotheses explaining the increase in microbial diversity in early HCC compared to cirrhosis, as it contrasts with the typical reduction in diversity seen in other disease contexts.

R: A sentence has been added at the end of the paragraph describing the alterations of GM in HCC patient cohorts. One reason of this discrepancy could be associated with the specific etiology (HBV) of this patient cohort with respect to the others. As suggested by the authors, another reason could be associated with overgrowth of pathogenic and pro-tumorigenic bacterial species in HCC patients; moreover, the authors argued that greater richness or diversity is not always a sign of a healthy gut microbiota.

  1. What is the mechanistic link between bacterial translocation and MT1B expression?

R: The authors reported an upregulation of MT1B in HCC compared to NAFLD and cirrhosis and identified a significant association of several bacterial genera with metallothionein MT1B, suggesting that bacterial translocation to the liver, arising from an increased perturbation of the gut barrier, determines a perturbation of gene expression in the liver altering the expression of genes involved in ROS scavenging. Since the authors do not report a mechanistic link between bacterial translocation and MT1B expression, we changed the sentence in text, accordingly. We thank the reviewer for this comment.

  1. Incorporate a table summarizing key microbial taxa and their roles in HCC progression, to enhance comprehension.

R: A Table summarizing key microbial taxa and their roles in HCC progression has been added to facilitate the comprehension. We also specified the etiology of HCC patients to highlight the differences between the studies.

  1. Line 232-give references in support of the statement.

R: A reference has been added, “Ref. N.52”.  

  1. Line 372: Add a period after “[55]”

R: A period has been added after “Ref. N.68” in the amended version of the manuscript.

  1. Figure 2: The name of the bacterium “Akkermansia munciniphila” in figure panel should be italicized.

R: The name of the bacterium “Akkermansia munciniphila” in figure has been italicized. We thank the reviewer for this correction.

  1. Use consistent terminology throughout ("gut microbiota" vs. "gut microbiome").

R: We apologize for the lack of consistency. The terminology has been revised in favor of the term “gut microbiome” wherever appropriate.

  1. Some mechanistic details of metabolites are mentioned such as the role of SCFAs in ROS production and apoptosis, but others are not explained in detail. Discuss how specific microbial metabolites or bile acids influence signaling pathways in depth.

R: As requested, we added the description of signaling pathways regulated by bile acids at the beginning of the related chapter (Ref. N. 54, 55). Because of reviewer 2's specific concern (point 1) about the excessive length of this review and because it is not the main topic of this review, we did not go into too much detail regarding the molecular aspects underneath BAs deregulation in CLD and cited other reviews that better explain this aspect. We thank the reviewer for this comment that allowed us to add information on BAs signaling.

  1. The role of isoallolithocholic acid in inducing immunosuppressive macrophages could be elaborated further to explain how this metabolite interacts with liver cells or pathways. Similarly, while discussing impact of DCA-producing bacteria on fibrosis progression, give clearer explanation of how these bacteria alter bile acid composition and affect liver function.

R: As requested a further explanation of the role of isoallolithocholic acid with liver-resident macrophages has been added. Two sentences and two new references (new Ref. N.86, 87) were also added to explain how these bacteria alter bile acid composition and affect liver function. We thank the reviewer for this comment allowing us to add further mechanistic information on GM-derived metabolites and liver damage.

  1. How might dietary habits or environmental factors contribute to geographic differences in microbiota composition?

R: Further investigations are needed to explain the unexpected variation in the composition of the microbiome of healthy individuals among different geographic area. In fact, it is difficult to explain whether the geographical effect is driven by host-specific factors or introduced by other ecological processes (new Ref. N. 83, 84). We changed that sentence accordingly and added a reference regarding the influence of nutritional components (e.g., fats) on gut microbiota composition in healthy individuals (new Ref. N. 82). We thank the reviewer for this comment allowing us to better clarify this issue.

  1. Discuss the safety and ethical considerations of FMT, especially in clinical settings.

R: The safety and ethical considerations of FMT have been discusses at the end of chapter 3.2 (new Ref. N. 122, 123). We thank the reviewer for this interesting comment.

  1. How does atorvastatin mediated GM modulation and the prevention of NAFLD-HCC relates to cholesterol metabolism and GM-derived metabolites?

R: An explanation regarding the mechanisms linking GM-derived metabolites to metabolic and proliferative effects in human hepatocytes and the role of atorvastatin in this process and in cholesterol metabolism has been added. We thank the reviewer for this comment allowing us to better clarify this aspect.

  1. Link preclinical findings to clinical implications, such as how probiotics or anti-cholesterol drugs could be integrated into current NAFLD management strategies.

R: A conclusive sentence regarding the clinical implications of probiotics (new Ref. N. 116) and statins (end of chapter 3.2) in the management of NAFLD patients has been added. Taking into considerations the results from preclinical studies and clinical trials, the use of statins in NAFLD patients may occur in the near future, whereas further studies on probiotic formulations are needed to obtain more consistent results in preventing NAFLD progression in humans.  

  1. Figure 2: “Ipocaloric diet” please correct the spelling.

R: The spelling of this word has been corrected. We thank the reviewer for this correction.

  1. Figure 3- Change “oxydative” to “oxidative”

R: The spelling of this word has been corrected. We thank the reviewer for this correction.

  1. Line 662- Replace "no therapeutic approaches to inhibit NAFLD progression are available in the clinics" with "no established therapeutic approaches are available to inhibit NAFLD progression in clinical practice."

R: The sentence has been changed accordingly. We thank the reviewer for this correction.

  1. Line 667, 670, 672, 673, 678: The name of bacterium “A. munciniphila” should be italicized.

R: The name of the bacterium “A. munciniphila” has been italicized in the text. We thank the reviewer for this correction.

  1. Line 679, 680, 684, 699: The name of bacterium “Bifidobacterium pseudolongum” should be italicized.

R: The name of the bacterium “B. pseudolongum” has been italicized in the text. We thank the reviewer for this correction.

  1. Line 684, 692: terms “in vitro” and “in vivo’ should be italicized.

R: The terms “in vitro” and “in vivo’ have been italicized in the text. We thank the reviewer for this correction.

  1. Line 688, 691: The name of bacterium “Lactobacillus acidophilus” should be italicized.

R: The name of the bacterium “Lactobacillus acidophilus” has been italicized in the text. We thank the reviewer for this correction.

  1. Line 698: The names of the bacteria “Mucispirillum, Desulfovibrio, Anaerotruncus” and the family “Desulfovibrionaceae” should be italicized.

R: The name of the bacteria “Mucispirillum, Desulfovibrio, Anaerotruncus” and the family “Desulfovibrionaceae” have been italicized in the text. We thank the reviewer for this correction.

  1. Figure 4: The name of the bacterium “Akkermansia munciniphila” in the figure caption should be italicized.

       R: The name of the bacterium “Akkermansia munciniphila” in the figure panel and caption has been italicized.

  1. Line 744- change “Authors” to “authors”.

      R: The word has been corrected.

  1. Discuss the challenges presented by GM modulation such as, variability in GM composition across individuals and populations, potential risks of probiotic supplementation like dysbiosis or overgrowth of certain taxa, feasibility of standardizing GM based interventions in a clinical setting.

R: In the conclusions section, we critically discussed all the concerns raised by the reviewer regarding the challenges of precision GM modulation, such as GM and host variability, potential risks of probiotic supplementation, and the difficulty of standardizing GM-based interventions in a clinical setting (new Ref. N. 151-154). We thank the reviewer for this critical comment.

  1. Define "objective response rate (ORR)" for non-specialist readers.

      R: The abbreviation ORR has been defined in the introduction chapter.

We really thank the reviewer for taking time to revise our manuscript and for his/her accurate comments, suggestions and corrections that allowed us to improve the quality of our manuscript and to add further mechanistic details. The tracked version of our manuscript is enclosed.

Reviewer 2 Report

Comments and Suggestions for Authors

In cells-3416283, Monti et al discuss the role of gut microbiome  in the pathogenesis of  hepatocellular carcinoma. The topic of this review is interesting and fits well the scope of Cells. The reviewer feels it can be accepted after some minor amendments.

(1) The review article  is too lengthy. It is not a pleasant process to read through. The authors should reduce its length and make it more attractive.

(2) The authors should discuss the impact of gender / race on gut microbiome.

(3) Is there any clinically applicable intervention to modulate gut microbiome?

Author Response

Reviewer 2

In cells-3416283, Monti et al discuss the role of gut microbiome in the pathogenesis of hepatocellular carcinoma. The topic of this review is interesting and fits well the scope of Cells. The reviewer feels it can be accepted after some minor amendments.

(1) The review article is too lengthy. It is not a pleasant process to read through. The authors should reduce its length and make it more attractive.

R: We thank the reviewer for taking time to revise our manuscript. To answer the reviewer’s request, we have eliminated two paragraphs in chapter 2.1 and 3. We have also added a Table at the end of chapter 2.2 to improve the comprehension and attractiveness of our manuscript. We apologize but we were not able to reduce the total length of our review because of all the concerns raised by reviewer 1, which have needed further explanations and mechanistic details. Nevertheless, we think that all these changes have improved the quality of our manuscript despite its length.

(2) The authors should discuss the impact of gender/race on gut 

R: As also requested by reviewer #1, in the conclusions section, we emphasized the need to consider microbiome-associated confounding factors (such as age, gender, ethnicity, lifestyle, geographic location, etc.), which may be responsible for the sometimes-inconsistent results between studies and may lead to differential treatment response.

(3) Is there any clinically applicable intervention to modulate gut microbiome?

R: As also requested by reviewer #1, in the conclusions section and in chapter 3.2, we discussed the clinical applications of gut microbiome modulation. In particular, fecal microbiota transplantation is an effective procedure used for the treatment of recurrent Clostridium difficile infection (CDI). Because of ethical and safety concerns, this practice cannot be used for any indication other than CDI without an ‘investigational new drug’. Regarding probiotic supplementation, we reported some inconsistencies across literature studies (chapter 3.2) for the treatment of NAFLD and we emphasized (conclusions section) the need to consider microbiome-host related basal variabilities, such as age, gender, genetics, microbiome composition, which may differently influence the response to probiotics in both healthy individuals and NAFLD or HCC patients. We highlighted that, despite promising in the preclinical setting, further studies and ad hoc clinical trials are needed before the application of gut microbiome modulation strategies in the clinics.

All the changes have been underlined in the tracked version of our manuscript.